# AVESFormer: Efficient Transformer Design for Real-Time Audio-Visual Segmentation

## Abstract

Recently, Transformer-based models have performed remarkably well in audio-visual segmentation (AVS) tasks. However, previous methods exhibit abnormal behavior and unsatisfactory results when using cross-attention. By analyzing attention maps, we identify two primary challenges in existing AVS models: 1) *attention dissipation,* caused by anomalous attention weights after Softmax over limited frames, and 2) *narrow attention patterns* in early decoder stages lead to inefficient utilization of attention mechanism. In this paper, we introduce *AVESFormer*, the first real-time audio-visual segmentation transformer that simultaneously achieves fast, efficient, and lightweight. Our model proposes an efficient, prompt query generator to rectify cross-attention behavior. Moreover, we propose an early focus (ELF) decoder, which enhances efficiency by incorporating convolution operations tailored for local feature extraction, thus reducing computational overhead. Extensive experiments demonstrate that AVESFormer effectively mitigates cross-attention issues, substantially improves attention utilization, and outperforms the previous state-of-the-art, achieving a superior trade-off between performance and speed. The code can be found in the supplementary material.

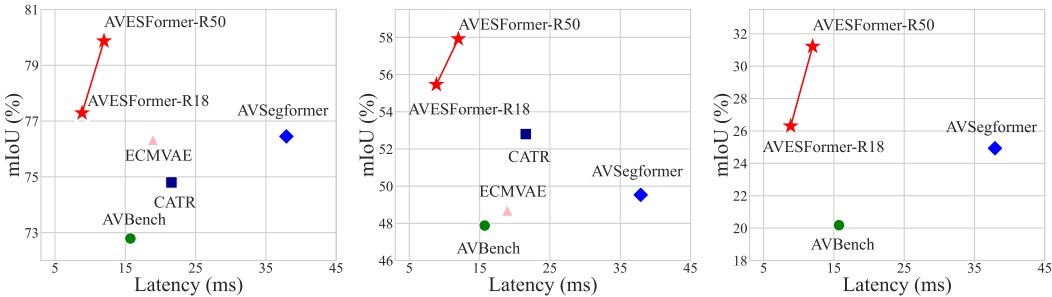

Figure 1: **mIoU (%) vs. Inference Latency (ms)** on S4 (left), MS3 (middle), and AVSS (right) compared with other popular methods. Latency is measured on a single Nvidia RTX 3090 GPU. AVESFormer achieves the best trade-off between mIoU and inference latency.

## 1 Introduction

Audio-Visual Segmentation (AVS) (Zhou et al., 2022) has emerged as a novel multi-modality task that plays a crucial role in robot sensing, video surveillance, and other scenarios. It aims to segment fine-grained pixel-level sounding objects with corresponding audio-visual modalities. However, existing AVS methods primarily focus on improving performance, often at a high cost of model size and computational overhead (Gao et al., 2024; Mao et al., 2023b; Liu et al., 2023b; Huang et al., 2023; Liu et al., 2023a; 2024a; Li et al., 2023b). Besides, default AVS setting directly processing $T$ frames at a time (Zhou et al., 2023; Li et al., 2023a) is also unfitted for immediate response. These drawbacks render them unsuitable for applications with real-time requirements.

Recently, transformer-based models have brought significant improvements to AVS (Gao et al., 2024; Yang et al., 2023; Huang et al., 2023; Li et al., 2023b; Liu et al., 2023b; 2024a; Li et al., 2023a). However, AVS models often rely on modified attention variants despite the prevalence of

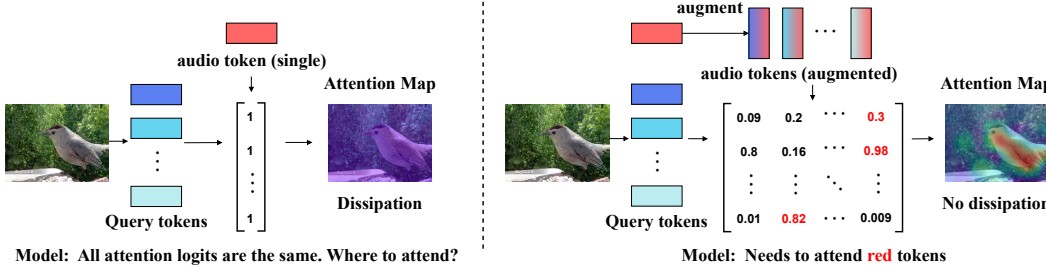

Figure 2: Illustration of attention dissipation. The cross-attention matrix fails to distinguish different tokens (left). One potential solution is to expand the audio feature into several tokens (right).

cross-attention for modality fusion within a single image in vision-language models (Li et al., 2022a; 2021; Luo et al., 2023). For instance, AVSegFormer (Gao et al., 2024) employs channel attention mixer (CHA) to guide visual channels with audio. However, CHA may be dominated by visual features and surpass audio representation (Chen et al., 2024). Chen et al. (2024) replaces Softmax in attention with Sigmoid, suggesting it could highlight critical regions. Stepping-Stones (Ma et al., 2024) proposes cosine similarity attention for audio guidance in audio-visual fusion. While these adaptations have shown some success, attention variants generally do not exhibit the same expressive capacity as the default mechanism (Tay et al., 2022). Therefore, a natural question arises: Why is AVS's conventional cross-attention fusion mechanism underutilized?

To this end, our studies start with the comprehensive observation and exploration of cross-attention. We characterize the attention probabilities and heatmaps within the cross-attention of AVS models. It reveals two critical issues behind them: (1) *Attention Dissipation*, a previously unexplored phenomenon, where cross-attention matrix vanishes in previous attempts, hindering them from distinguishing audio-visual corresponding regions, as illustrated in Figure 2. It erupts intensely in an improper attention configuration and real-time AVS scenario. (2) *Narrow Attention Pattern*, an inefficient heatmap pattern in cross-attention map after solving attention dissipation. Attention maps at early decoder stages tend to capture short-term local correlation features, leading to undesired low utiliza-

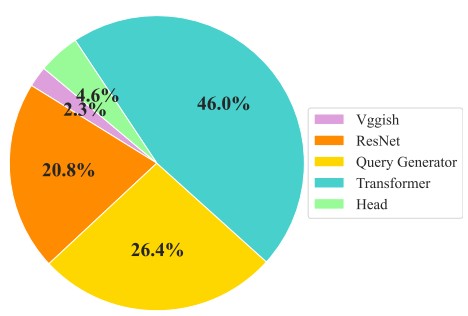

Figure 3: Runtime profiling of the AVSeg-Former (Gao et al., 2024).

tion of attention. These limitations not only obstruct the formation of long-range dependencies but also contribute to the inference runtime bottleneck. As depicted in Figure 3, the runtime proportion of the transformer, including the query generator, can exceed 70% of the total.

In this work, we introduce AVESFormer, an *Audio-Visual Efficient Segmentation Transformer* for real-time AVS, seeking to refine the cross-attention mechanism through theoretical insights and enhanced utilization of multi-modality features. First, we find that attention dissipation is derived from the peculiar shape of the attention weights under the Softmax function and is prominently reflected in real-time AVS and single-frame modality fusion. To address this issue, *Prompt Query Generator* (PQG) is adopted to process audio features as a prompt, rebuilding distinguishing ability and effectively eliminating attention dissipation. A novel *Early Focus* (ELF) decoder is proposed for narrow attention patterns. Specifically, convolution is introduced in the early transformer decoder stages, enabling more effective local feature interaction in contrast to the wasted inefficient attention while reducing the computational cost of the latter.

We evaluate our AVESFormer on S4, MS3, and AVSS tasks on challenging AVSBench dataset (Zhou et al., 2023). As shown in Figure 1, comprehensive experiments show that AVESFormer achieves state-of-the-art performance-latency trade-off. Furthermore, we also present that AVESFormer outperforms previous transformer-based model (Gao et al., 2024) by +3.4% on S4, +8.4% on MS3 and +6.3% on AVSS while using 20% less parameters and 3× speed-up.

## 2 RELATED WORK

**Real-Time Audio-Visual Segmentation.** Audio-Visual Segmentation (AVS) is a more fine-grained and complicated task than sound source localization (SSL) (Chen et al., 2021a; Hu et al., 2020; Qian et al., 2020b) as it aims to locate the sounding object and show pixel-level predictions. However, few research works focus on real-time scenario where only 1 frame is given at a time instead of $T$ frames. AVSBench (Zhou et al., 2022) is the first to propose audio-visual segmentation benchmark, introducing temporal pixel-wise audio-visual interaction (TPAVI) module to facilitate interaction between audio-visual information. AVSegFormer (Gao et al., 2024) is the first to develop a novel transformer architecture for AVS. They introduce audio queries into the transformer decoder to attend to corresponding visual features. CATR (Li et al., 2023a) performs bidirectional combinatorial dependence fusion to fully enhance spatial-temporal dependencies. (Chen et al., 2024) incorporates contrastive loss into audio-visual semantic segmentation with positive and negative pairs and uses larger resolution with extra data to reach higher performance.

Nevertheless, these methods encounter issues when dealing with single frame image and audio, making them hardly work for real-time scenario. In detail, many research works meet a failure case in cross attention with vision as query and audio as key and value. We call this failure case **Attention Dissipation**. AVSegFormer (Gao et al., 2024) fails to deliver satisfactory results when firstly trying Cross-Attention Mixer (CRA). (Chen et al., 2024) generates a plain attention map when visualizing Softmax attention map in their work. To tackle this problem, researchers propose different cross attention variants to amend it. TPAVI in AVSBench performs modality fusion by the dot-product of vision and audio, which can be regarded as a linear attention. AVSegFormer employs a query generator and perform channel attention to expand audio features and to avoid audio as key and value. Chen et al. (2024) proposes Sigmoid attention to replace Softmax. Stepping-Stones (Ma et al., 2024) proposes Adaptive Audio Query Generator, which obtains audio-conditioned query by cosine similarity to enrich audio features. Although many alternative attention methods are proposed, the underlying problem still remains unexplored. These attention variants can achieve some results, but their expression ability is still not sufficient to match default attention(Tay et al., 2022). Therefore, it is necessary to amend the behaviour of cross attention.

**Efficient Vision Transformer.** ViT (Dosovitskiy et al., 2020) and its variants (Liu et al., 2021; Touvron et al., 2021; Wang et al., 2022) have demonstrated significant improvements in computer vision. However the high computational cost makes them inferior to CNN in real-time inference scenario. To mitigate this gap, previous works attempt to design more efficient architectures to reduce computational burden. MobileViT (Mehta & Rastegari, 2021) combines CNN and ViT by integrating global feature fusion of transformer in CNN. MobileFormer (Chen et al., 2022) bridges MobileNet (Howard et al., 2017) and ViT in a parallel design to leverage advantages from both architectures. EfficientFormer (Li et al., 2022b) finds insufficient operations in transformer and slims the model size in a latency-driven manner. LVT (Xiao et al., 2021) adopts dilated convolution in attention mechanisms to enhance model performance and efficiency. LIT (Pan et al., 2022) gives a more detailed analysis of self-attention heads and applies MLP to build local dependencies. EfficientViT (Cai et al., 2022) proposed to aggregate multi-scale features via small-kernel convolutions. These methods have made contributions to the development of efficient ViT architectures. We benefit greatly from their contributions to the analysis of AVS tasks.

## 3 REVISITING AVS UNDER REAL-TIME SCENARIO

### 3.1 PRELIMINARIES

This paper considers real-time audio-visual segmentation, which is different from common AVS task settings. Traditional AVS tasks deal with a clip of video frames, which contains $T$ visual frames $x_{visual} \in \mathbb{R}^{T \times 3 \times H \times W}$, and corresponding audio signals $x_{audio} \in \mathbb{R}^{T \times D}$, where $H$ and $W$ are the height and width of the image and $D$ is the audio dimension.

However, it's impractical for real-time inference on a whole bunch of $T$ frames at a time. Users expect an immediate response as a single input is given instead of waiting for the entire $T$ frames to be processed together. Meanwhile, the limited memory of edge devices is insufficient to handle the entire video clip. Therefore, this paper aims at a more practical AVS scenario, called **real-time**

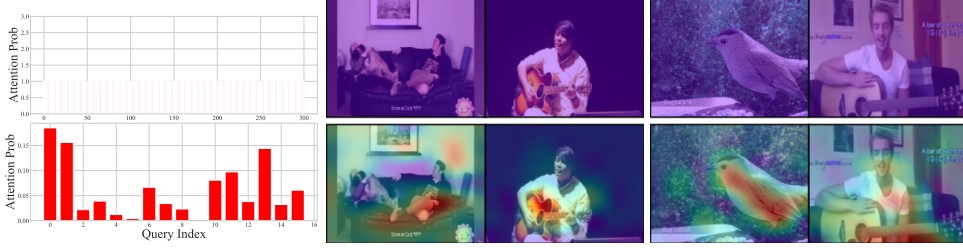

(a) Prob in query generator      (b) Single frame heatmap      (c) Heatmap when set $T = 1$

Figure 4: (a) Upper: Attention probabilities assigned to each audio query in query generator (Gao et al., 2024), leading to a plain distribution. Lower: Attention prob by our AVESFormer, without dissipation. (b) and (c) Upper: the plain heatmap in single frame fusion and real-time scenario. Lower: the amended heatmap in AVESFormer.

**AVS**, where only one single frame is segmented, and the time dimension is forced to $T = 1$. Only one image and a piece of audio signal are given for one segmentation mask.

Specifically, initially, audio-visual features are extracted by corresponding backbones. For input image $x_{visual} \in \mathbb{R}^{3 \times H \times W}$, hierarchical visual features $\mathcal{F}_{visual}$ are extracted by visual backbone. Meanwhile, the audio signal is resampled to yield a 16kHz mono output $A_{mono} \in \mathbb{R}^{N_{samples} \times 96 \times 64}$, where $N_{samples}$ stands for the number of sampling points. Then, $A_{mono}$ is converted into Mel-spectrum $A_{mel} \in \mathbb{R}^{96 \times 64}$ by short-time Fourier transform. Finally we put $A_{mel}$ into audio backbone to extract features, denoted as $\mathcal{F}_{audio} \in \mathbb{R}^{1 \times D}$, where $D$ is the audio feature dimension. The goal of AVS is to segment the corresponding sounding visual object region $\mathcal{M} \in \mathbb{R}^{N_{class} \times H \times W}$ given the audio sounding signal, where $N_{class}$ is the number of class labels.

### 3.2 MOTIVATION OBSERVATIONS

In real-time AVS, visual feature $\mathcal{F}_{visual} \in \mathbb{R}^{c \times h \times w}$ and audio feature $\mathcal{F}_{audio} \in \mathbb{R}^{1 \times c}$ are given at the same moment. The former is usually split into patches $\mathcal{P}_{visual} \in \mathbb{R}^{N \times c}$ where $N = h \times w$ for attention operation. The common approach directly performs cross-attention, as shown on the left panel of Figure 2. Let us denote $q_i, k, v \in \mathbb{R}^{1 \times c}$ as row vectors for $i \in [1, 2, \ldots, N]$, with $\mathcal{P}_{visual} = [q_i]_{N \times c}$ and $\mathcal{F}_{audio} = k = v$. The cross-attention fusion can be represented as follows:

$$\mathcal{O} = \text{Softmax}(\mathcal{P}_{visual}\mathcal{F}_{audio}^T)\mathcal{F}_{audio}, \tag{1}$$

$$o_i = \sum_j a_{i,j} v_j = \frac{\sum_j e^{q_i k_j^T} v_j}{\sum_j e^{q_i k_j^T}}, \tag{2}$$

where $\mathcal{O} = [o_i] \in \mathbb{R}^{N \times c}$ and $j$ stands for the row index of $\mathcal{F}_{audio}$. The scale factor $\sqrt{d}$ in Softmax as well as linear transformation matrices of $W^Q$, $W^K$ and $W^V$ (Vaswani et al., 2017) are omitted for the sake of simplicity without affecting the conclusion.

However, $\mathcal{F}_{audio}$ is an 1-dimensional vector, which makes $k_j = k$ and $v_j = v$. Based on this hypothesis, we substitute $j = 1$ into Equation (2) to obtain:

$$o_i = \frac{e^{q_i k^T} v}{e^{q_i k^T}} = v. \tag{3}$$

The final output of cross-attention fusion can be written as:

$$\mathcal{O} = \text{Softmax}(\{q_i k^T\}_{ij})v = \mathbf{1}_{N \times 1}\mathcal{F}_{audio} = [\mathcal{F}_{audio}]_{N \times c}. \tag{4}$$

From Equation (4), the cross-attention fusion turns into a simple replication of the audio feature, as illustrated on the right panel of Figure 2. The phenomenon revealed in Equation (4), termed **Attention Dissipation**, significantly harms the capability of distributing attention on multi-modality representation, thus constraining the effectiveness of the attention mechanism (Gao et al., 2024; Chen et al., 2024). See Appendix A.1.1 for more proof details.

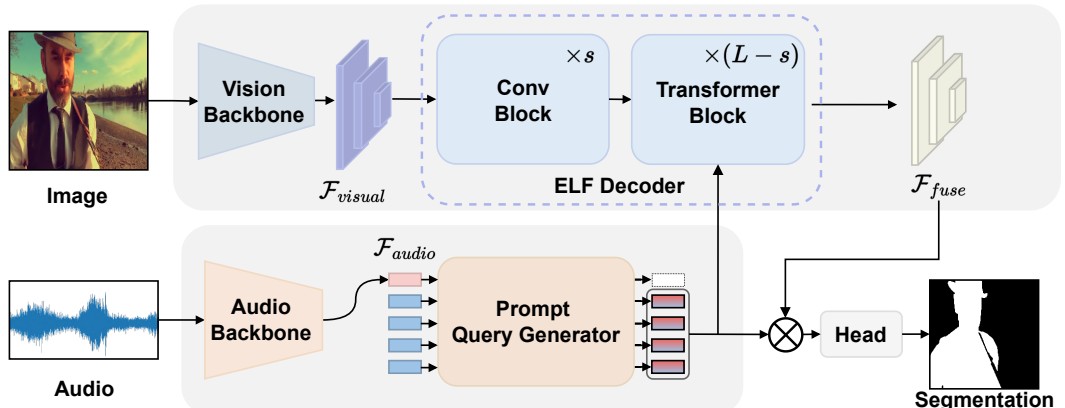

Figure 5: The overview of AVESFormer. The prompt query generator addresses attention dissipation by inserting the audio feature on top of learnable parameters to generate audio-conditioned queries. The ELF decoder processes local features using convolution blocks in the early stages.

Furthermore, attention dissipation appears in various situations, as shown in Figure 4. It leads to the failure of the Cross-Attention Mixer (CRA) tried by Gao et al. (2024). But this phenomenon still remains in their query generator, where cross-attention is performed on the individual audio features as key, as shown in Figure 4(a). Chen et al. (2024) observes a plain Softmax attention map in their visualization, as depicted by Figure 4(b), but doesn't conduct further exploration. Moreover, temporal audio-visual fusion under real-time AVS also appears attention dissipation (Li et al., 2023a; Liu et al., 2023b), as shown in Figure 4(c).

# 4 METHOD

We now aim to perform proper cross-attention fusion for real-time AVS. Concretely, we are given a single visual frame $x_{visual} \in \mathbb{R}^{3 \times H \times W}$, and a raw audio signal $A_{mono} \in \mathbb{R}^{N_{samples} \times 96 \times 64}$. Our goal is to learn a model that could successfully predict the segmentation mask $\mathcal{M}$. We elaborate on the detailed architecture and components of the proposed AVESFormer as shown in Figure 5.

## 4.1 PROMPT QUERY GENERATOR

Previous query generator module with default cross-attention, e.g., AVSegFormer Gao et al. (2024), tries to generate audio-conditioned features by modeling $p(z|\mathcal{F}_{audio})$, where $z$ is the learnable queries, to produce the audio queries related to current audio signals. The scaled-dot-product attention measures the relevance. However, this method fails because of the attention dissipation of learnable queries, such as $Q$, and each individual audio feature, such as $K$ and $V$.

Similarly focused on obtaining audio-conditioned queries via $p(z|\mathcal{F}_{audio})$, we propose a novel prompt query generator (PQG), as depicted in Figure 6. The audio feature in a single frame is regarded as a **prompt** (Liu et al., 2023c) and concatenated on the head of a set of learnable queries $Q_{learn} \in \mathbb{R}^{N_q \times D}$:

$$Q^{\dagger} = [\mathcal{F}_{audio}|Q_{learn}] \in \mathbb{R}^{(N_q+1) \times D}, \quad (5)$$

where $[\cdot|\cdot]$ denotes concatenation and $N_q$ denotes query number. Then, PQG calculates relevance between learnable queries and audio features by self-attention. Each learnable query may convey part of the related information from the original audio feature, and overall, they inherit the information.

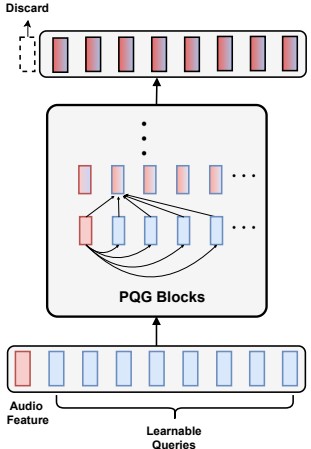

Figure 6: Illustration of the prompt query generator.

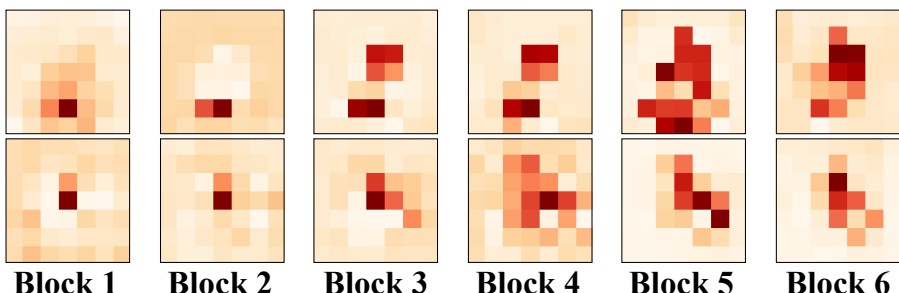

| **Block 1** | **Block 2** | **Block 3** | **Block 4** | **Block 5** | **Block 6** |

Figure 7: Attention probabilities of different blocks in fully transformer decoder. Each map shows the attention probability of the audio query to all visual patches. Maps are averaged along all heads and queries. Each row indicates a test sample. Dark red indicates higher attention probability, and early orange indicates lower attention probability.

Finally, the original audio feature is discarded at the output to obtain $\mathcal{F}_{gen} \in \mathbb{R}^{N_q \times D}$. It is important to note that PQG serves as an effective approach for modeling $p(z|\mathcal{F}_{audio})$. While preserving the information from the original audio feature, the generated audio features also avoid attention dissipation within itself and the following operation.

## 4.2 EARLY FOCUS DECODER

Our approach is based on the audio-visual cross-attention patterns, as shown in Figure 7. In the early stages, audio features generate narrow local responses on attention maps. In the early stages, audio features generate narrow local responses on attention maps. As it goes deeper, the attention region enlarges gradually and, in the end, forms shaped regions suitable for segmentation. Therefore, we propose a novel early focus (ELF) decoder. Since the early stage primarily captures local patterns, attention to high computational cost is replaced by convolution to capture local semantics. In early decoder stage $l$, visual feature $\mathcal{F}_{visual}$ is processed by convolution:

$$\mathcal{F}_{visual}^{l+1} = \text{LN}(\mathcal{F}_{visual}^l + \text{Conv}(\mathcal{F}_{visual}^l)), \tag{6}$$

where LN denotes LayerNorm (Ba et al., 2016). In deeper stages, we split $\mathcal{F}_{visual}$ into visual patches $\mathcal{P}_{visual}$ (Dosovitskiy et al., 2020) to perform cross-attention with $\mathcal{F}_{gen}$ from PQG:

$$\mathcal{P}_{visual}^{l+1} = \text{LN}(\mathcal{P}_{visual}^l + \text{CA}(\mathcal{P}_{visual}^l, \mathcal{F}_{gen}, \mathcal{F}_{gen})), \tag{7}$$

where CA denotes multi-head cross-attention and $\text{CA}(Q, K, V) = \text{Softmax}(QK^T)V$. The ELF decoder eliminates the computational burden brought by wasted attention operations but still maintains the original module function to extract local features.

## 5 EXPERIMENTS

**Dataset.** We evaluate our method on the AVSBench dataset (Zhou et al., 2022; 2023), which is composed of AVSBench-Object and AVSBench-Semantic. AVSBench-Object is designed for audio-visual segmentation tasks with pixel-level annotations with two subsets: single sound source segmentation (S4) subset and multiple sound source segmentation (MS3) subset. AVSBench-Semantic is an expanded version of AVSBench-Object, providing additional semantic masks to facilitate audio-visual semantic segmentation (AVSS). See Appendix A.2.1 for more experimental details.

**Implementation Details.** Our model is trained on NVIDIA RTX 3090 GPU. From the aspect of real-time inference, we employ ResNet-50 and ResNet-18 (He et al., 2016) pre-trained on ImageNet (Russakovsky et al., 2015) as our visual backbones. Considering Pyramid Vision Transformer (PVT-v2) (Wang et al., 2022) is unsuitable for real-time applications, we do not adopt it as the visual backbone. We employ Vggish (Hershey et al., 2017) pre-trained on AudioSet (Gemmeke et al., 2017) to encode audio input. Jaccard index $\mathcal{J}$ and F-score $\mathcal{F}$ are adopted as evaluation metrics. See Appendix A.2.1 for more experimental details.

Table 1: Comparison with state-of-the-art methods on the S4, MS3 benchmark. The evaluation metrics are Jaccard index and F-score.

| Method | Backbone | S4 | | MS3 | |
|---|---|---|---|---|---|
| | | $\mathcal{J}$ | $\mathcal{F}$ | $\mathcal{J}$ | $\mathcal{F}$ |
| LVS (Chen et al., 2021b) | ResNet-18 | 38.0 | 51.0 | 29.5 | 33.0 |
| MSSL (Qian et al., 2020a) | ResNet-18 | 44.9 | 66.3 | 26.1 | 36.3 |
| 3DC (Mahadevan et al., 2020) | ResNet-152 | 57.1 | 75.9 | 36.9 | 50.3 |
| SST (Duke et al., 2021) | ResNet-101 | 66.3 | 80.1 | 42.6 | 57.2 |
| iGAN (Mao et al., 2021) | Swin-T | 61.6 | 77.8 | 42.9 | 54.4 |
| LGVT (Zhang et al., 2021) | Swin-T | 74.9 | 87.3 | 40.7 | 59.3 |
| AVSBench (Zhou et al., 2022) | | 72.8 | 84.8 | 47.9 | 57.8 |
| CATR (Li et al., 2023a) | | 74.8 | 86.6 | 52.8 | 65.3 |
| DiffusionAVS (Mao et al., 2023a) | | 75.8 | 86.9 | 49.8 | 62.1 |
| ECMVAE (Mao et al., 2023b) | | 76.3 | 86.5 | 48.7 | 60.7 |
| AuTR (Liu et al., 2023b) | | 75.0 | 85.2 | 49.4 | 61.2 |
| AQFormer (Huang et al., 2023) | ResNet-50 | 77.0 | 86.4 | 55.7 | 66.9 |
| AVSC (Liu et al., 2023a) | | 77.0 | 85.2 | 49.6 | 61.5 |
| AVSegFormer (Gao et al., 2024) | | 76.5 | 85.9 | 49.5 | 62.8 |
| AVSBG (Hao et al., 2024) | | 74.1 | 85.4 | 45.0 | 56.8 |
| BAVS (Liu et al., 2024a) | | 78.0 | 85.3 | 50.2 | 62.4 |
| UFE (Liu et al., 2024b) | | 79.0 | 87.5 | 55.9 | 64.5 |
| MUTR (Yan et al., 2024) | | 78.6 | 87.3 | 57.0 | 66.1 |
| AVESFormer (ours) | ResNet-18 | 77.3 | 87.5 | 55.5 | 65.1 |
| | ResNet-50 | **79.9** | **89.1** | **57.9** | **68.7** |

## 5.1 MAIN RESULTS

Comprehensive experiments have been conducted on AVSBench-Object and AVSBench-Semantic datasets alongside other methods. As shown in Table 1 and Table 2. Model parameter counts and inference latency is presented in Table 3. Our AVESFormer exhibits the state-of-the-art performance-speed trade-off among all models. Specifically, AVESFormer surpasses previous methods w.r.t. mIoU by 79.9% on the S4 subset, 57.9% on the MS3 subset and 31.2% on the AVSS subset, respectively. Figure 1 illustrates that the inference speed of AVESFormer exceeds previous methods with the ResNet-50 backbone by large margins. In summary, these results demonstrate the advantages of AVESFormer in terms of performance, speed, and model size.

Table 2: Comparison with state-of-the-art methods on the AVSS benchmark. The evaluation metrics are Jaccard index and F-score.

| Method | Backbone | AVSS | |
|---|---|---|---|
| | | $\mathcal{J}$ | $\mathcal{F}$ |
| 3DC (Mahadevan et al., 2020) | ResNet-152 | 17.3 | 21.6 |
| AOT (Yang et al., 2021) | Swin-B | 25.4 | 31.0 |
| AVSBench (Zhou et al., 2022) | | 20.2 | 25.2 |
| AVSegFormer (Gao et al., 2024) | ResNet-50 | 24.9 | 29.3 |
| BAVS (Liu et al., 2024a) | | 24.7 | 29.6 |
| AVESFormer (ours) | ResNet-18 | 26.3 | 31.8 |
| | ResNet-50 | **31.2** | **36.8** |

Table 3: Comparison with state-of-the-art methods on parameter counts and latency. #Params refers to the number of parameters. Latency is reported on a single NVIDIA RTX 3090 GPU. * means the parameters of audio backbone Vggish (Hershey et al., 2017) are included.

| Method | Backbone | #Params* (M) | Latency (ms) |
|---|---|---|---|
| AVSBench (Zhou et al., 2022) | | 163 | 15.7 |
| CATR (Li et al., 2023a) | ResNet-50 | 177 | 21.6 |
| ECMVAE (Mao et al., 2023b) | | 162 | 18.9 |
| AVSegFormer (Gao et al., 2024) | | 151 | 37.9 |
| AVESFormer (ours) | ResNet-18 | **108** | **8.8** |
| | ResNet-50 | 127 | 12.0 |

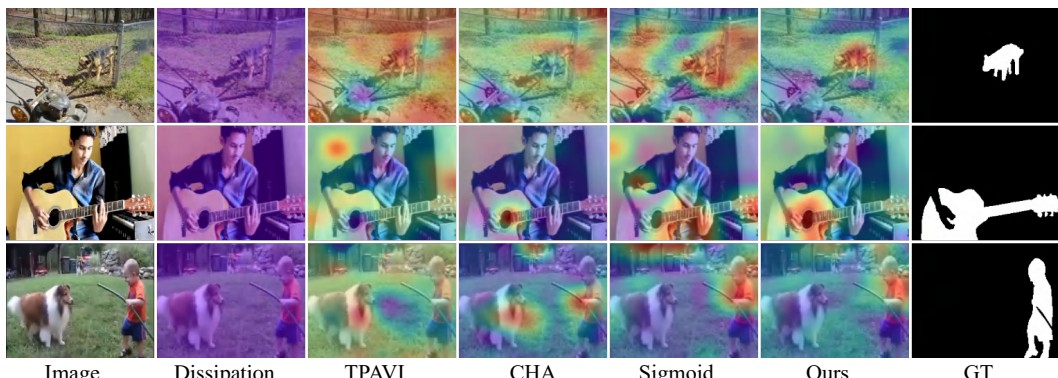

| Image | Dissipation | TPAVI | CHA | Sigmoid | Ours | GT |

Figure 8: Visualization of attention maps, including cross-attention with attention dissipation, TPAVI (Zhou et al., 2022), channel attention mixer (CHA) (Gao et al., 2024), sigmoid attention (Chen et al., 2024) and our ELF decoder. Each map shows the correlation between audio queries and visual patches. Red indicates a higher attention score, while blue indicates a lower one.

## 5.2 HANDLING ATTENTION DISSIPATION

**Effectiveness of Prompt Query Generator** To verify the effectiveness of PQG, we remove it to fuse modality with raw, unprocessed audio features. Additionally, the original query generator (QG) proposed by Gao et al. (2024) and an optional bias query generator (BQG) are also included. The ordinary query generator follows default settings with 6 layers and 300 queries. The bias query generator replicates the audio query and adds a learnable bias term. As shown in Table 4, PQG treats the audio feature as a prompt and cleverly addresses dissipation to avoid attention dissipation, yielding more improvements than the bias query generator.

Table 4: Effect of PQG. PQG overcomes attention dissipation to gain more improvements.

| Method | S4 $\mathcal{J}$ | S4 $\mathcal{F}$ | MS3 $\mathcal{J}$ | MS3 $\mathcal{F}$ |
|---|---|---|---|---|
| w/o QG | 75.9 | 87.1 | 50.0 | 61.9 |
| w/ QG | 78.5 | 88.7 | 50.0 | 61.7 |
| w/ BQG | 75.9 | 87.1 | 49.6 | 60.0 |
| **w/ PQG** | **79.9** | **89.1** | **57.9** | **68.7** |

Table 5: Performance of different fusion strategies. After fixing attention dissipation, cross-attention fusion works still best.

| Method | S4 $\mathcal{J}$ | S4 $\mathcal{F}$ | MS3 $\mathcal{J}$ | MS3 $\mathcal{F}$ |
|---|---|---|---|---|
| dissipation | 79.2 | 88.1 | 47.1 | 60.9 |
| w/ TPAVI | 79.6 | 88.7 | 55.4 | 65.4 |
| w/ CHA | 79.6 | 88.6 | 55.7 | 65.8 |
| w/ sigmoid | 78.4 | 88.6 | 55.3 | 62.0 |
| **w/ cross attn** | **79.9** | **89.1** | **57.9** | **68.7** |

Table 6: Performance of AVSegFormer (Gao et al., 2024) and AVESFormer with QG and PQG.

| Model w/ Method | S4 | | MS3 | | #Params | Latency |
|---|---|---|---|---|---|---|
| | $\mathcal{J}$ | $\mathcal{F}$ | $\mathcal{J}$ | $\mathcal{F}$ | (M) | (ms) |
| AVSegFormer w/ QG | 76.5 | 85.9 | 49.5 | 62.8 | 151 | 37.9 |
| AVSegFormer w/ PQG | 77.4 | 86.9 | 56.0 | 67.7 | 144 | 32.5 |
| AVESformer w/ QG | 76.5 | 85.9 | 49.5 | 62.8 | 131 | 17.9 |
| **AVESformer w/ PQG** | **79.9** | **89.1** | **57.9** | **68.7** | **127** | **12.0** |

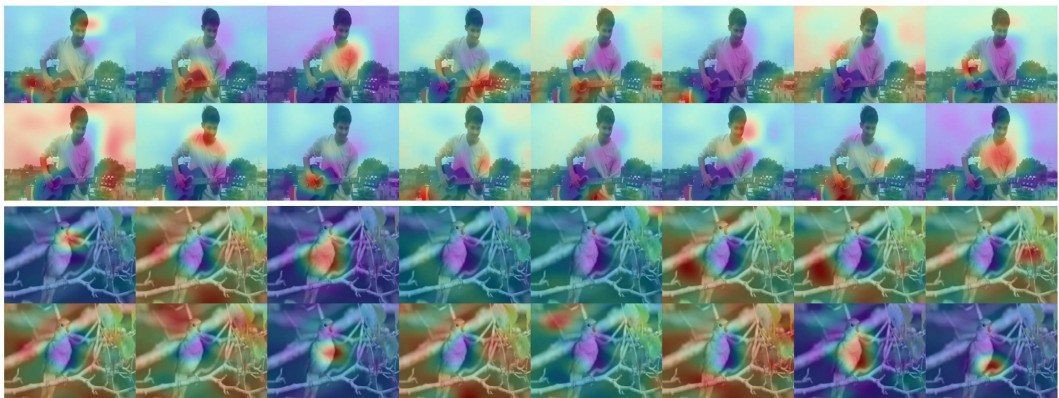

Figure 9: Visualization of attention maps by each audio query in PQG. Red indicates a higher attention score, while blue indicates a lower one.

**Intrinsic nature of PQG**    Attention maps of individual queries in PQG are visualized in Figure 9 to analyze its functionality. For a given frame, certain audio queries attend to the corresponding sounding object, while others may focus on the background. Each audio query captures distinct semantic features: some attend to specific parts of the sounding object, while others capture the entire object. Across different frames, queries adapt by attending to different objects. For instance, a query might focus on the sounding object in one frame but shift attention to the background in a different context. This demonstrates PQG's ability to effectively capture diverse semantic information in various audio-conditioned scenarios.

**Fusion Strategy.**    Furthermore, cross-attention fusion after addressing attention dissipation compared to other fusion strategies is investigated. Including a) cross-attention under attention dissipation, b) TPAVI by Zhou et al. (2022), c) CHA by Gao et al. (2024), d) sigmoid by Chen et al. (2024). Results are shown in Table 5. After addressing attention dissipation, cross-attention emerges as the optimal choice, demonstrating the most distinguishing representation ability. Figure 8 shows the attention map visualizations of different fusion strategies.

**Influence with Plug and Play PQG.**    Furthermore, PQG can be integrated into other models such as AVSegFormer (Gao et al., 2024), as shown in Table 6. On MS3, where the audio distinguishing capability is crucial due to the presence of multiple sound sources within an image, PQG demonstrates substantial improvement (+6.5% mIoU) when applied to AVSegFormer.

## 5.3    HYPERPARAMETERS AND ABLATION STUDIES ON AVESFORMER

**Training Setup.**    We provide ablation results with AVESFormer. To make quick evaluations, we adopt ResNet-50 as the backbone and perform extensive experiments on the S4 and MS3 sub-tasks. Other training settings remain consistent with Section 5.

**ELF Decoder.**    We analyze the influence of convolution at different stages of the ELF decoder. As shown in Table 7, "C" denotes convolution, and "T" denotes transformer. "Stage" indicates the

Table 7: Impact of the convolution blocks at different stages. We show model performance with different convolution insertion stages.

| Stage | S4 | | MS3 | | AVSS | | Latency |
|---|---|---|---|---|---|---|---|
| | $\mathcal{J}$ | $\mathcal{F}$ | $\mathcal{J}$ | $\mathcal{F}$ | $\mathcal{J}$ | $\mathcal{F}$ | (ms) |
| T-T-T | 77.3 | 87.6 | 56.2 | 66.6 | 30.7 | 35.1 | 14.9 |
| **C-T-T** | **79.9** | **89.1** | **57.9** | **68.7** | **31.2** | **36.8** | 12.0 |
| T-C-T | 77.6 | 88.0 | 56.5 | 67.3 | 29.3 | 35.1 | 12.1 |
| T-T-C | 77.1 | 88.3 | 55.2 | 67.3 | 31.0 | 36.4 | **11.8** |

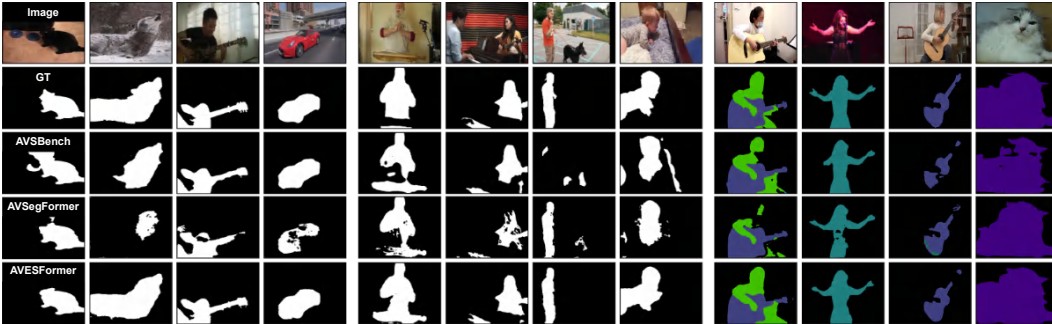

Figure 10: Visualization of segmentation predictions on S4 (left), MS3 (middle) and AVSS (right) Dataset with AVSBench (Zhou et al., 2022) and AVSegFormer (Gao et al., 2024).

insertion stage of convolution, with three options listed: early (C-T-T), middle (T-C-T), and deep (T-T-C). Additionally, a pure transformer decoder (T-T-T) is included. As convolution blocks are moved deeper, the mIoU drops by 2.81% on S4 and 2.73% on MS3. This decline can be attributed to the fact that early layers primarily generate local responses. In contrast, deeper layers facilitate high-level interactions between audio-visual modalities, which are essential for AVS tasks.

**Number of Queries.** Table 8 presents the performance of AVESFormer trained with varying numbers of quires of PQG in AVS-Bench. The experiments span query counts from 8 to 256 with a scale factor of 2. Notably, utilizing 16 queries performs best across S4 and MS3. This suggests that even though there are a number of sounding object categories, a large number of queries may not be necessary. A few queries in AVESFormer are adequate for learning distinguishing audio features.

Table 8: Performance of different number of queries in PQG.

| # of queries | S4 | | MS3 | |
|---|---|---|---|---|
| | $\mathcal{J}$ | $\mathcal{F}$ | $\mathcal{J}$ | $\mathcal{F}$ |
| 8 | 79.3 | 88.9 | 55.8 | 66.0 |
| **16** | **79.9** | **89.1** | **57.9** | **68.7** |
| 32 | 79.4 | 88.9 | 56.2 | 66.6 |
| 64 | 79.1 | 88.9 | 55.8 | 67.0 |
| 128 | 79.0 | 88.8 | 56.0 | 67.4 |
| 256 | 79.3 | 89.0 | 57.3 | 67.8 |

**Qualitative Analysis.** Visualizations of AVESFormer compared with those of AVSBench (Zhou et al., 2022) and AVSegFormer (Gao et al., 2024) are depicted in Figure 10. Our AVESFormer overcomes critical attention dissipation and makes more sophisticated visualization and segmentation performance. See Appendix A.3.1 for more visualizations.

## 6 CONCLUSION

In this paper, we analyze the attention dissipation phenomenon and inefficient transformer decoder. Based on these findings, we introduce AVESFormer, the first transformer-based real-time AVS model. Experimental results demonstrate that AVESFormer achieves the new state-of-the-art performance-speed trade-off. We hope our method provides insights into new architecture design not only in AVS tasks but also in various multi-modality scenarios.

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

# A APPENDIX

## A.1 ATTENTION DISSIPATION

### A.1.1 PROOF ON ATTENTION DISSIPATION

As discussed in Sec. 3.2, a brief explanation of attention dissipation is given. Now, we will provide more detailed proof of this phenomenon.

As commonly practised in AVS tasks, visual features are extracted from the visual backbone to get $\mathcal{F}_{visual} \in \mathbb{R}^{c \times h \times w}$ of one frame. Then we patchify the visual feature into $\mathcal{P}_{visual} \in \mathbb{R}^{N \times c}$ where $N = h \times w$. Meanwhile, audio signals within one frame are input into the audio backbone to form $\mathcal{F}_{audio} \in \mathbb{R}^{1 \times c}$. Note that since we only consider one frame at a time in real-time scenario, the sequence length of the audio feature is equal to 1. We cannot omit the sequence length dimension because we should keep this shape to perform matrix multiplication in the attention mechanism.

Consequently, the modality fusion process is performed originally by cross attention, where visual patches are query while the audio feature is key and value:

$$O = \text{Softmax}(\mathcal{P}_{visual}\mathcal{F}_{audio}^T)\mathcal{F}_{audio} \in \mathbb{R}^{N \times c}, \tag{8}$$

where

$$\mathcal{P}_{visual} = \begin{bmatrix} q_1 \\ q_2 \\ \vdots \\ q_N \end{bmatrix}, \tag{9}$$

$$q_i \in \mathbb{R}^{1 \times c}, \quad i \in [1, 2, \ldots, N], \tag{10}$$

$$\mathcal{F}_{audio} = k = v \in \mathbb{R}^{1 \times c}. \tag{11}$$

The attention logit matrix $\mathcal{A}$ can be written as:

$$\mathcal{A} = \mathcal{P}_{visual}\mathcal{F}_{audio}^T = \begin{bmatrix} q_1 \\ q_2 \\ \vdots \\ q_N \end{bmatrix} k^T = \begin{bmatrix} q_1 k^T \\ q_2 k^T \\ \vdots \\ q_N k^T \end{bmatrix} \in \mathbb{R}^{N \times 1}, \tag{12}$$

where

$$q_i k^T \in \mathbb{R}, \quad i \in [1, 2, \ldots, N]. \tag{13}$$

Softmax is calculated along the row vector on attention matrix $\mathcal{A}$ to get attention probability matrix $\mathcal{P}$:

$$\mathcal{P} = \text{Softmax}(\mathcal{A})|_{\text{row}} = \begin{bmatrix} e^{q_1 k^T}/\sum e^{q_1 k^T} \\ e^{q_2 k^T}/\sum e^{q_2 k^T} \\ \vdots \\ e^{q_N k^T}/\sum e^{q_N k^T} \end{bmatrix} = \begin{bmatrix} e^{q_1 k^T}/e^{q_1 k^T} \\ e^{q_2 k^T}/e^{q_2 k^T} \\ \vdots \\ e^{q_N k^T}/e^{q_N k^T} \end{bmatrix} = \begin{bmatrix} 1 \\ 1 \\ \vdots \\ 1 \end{bmatrix} = \mathbf{1}_{N \times 1}. \tag{14}$$

Finally the output $\mathcal{O}$ becomes a simply replication of value matrix:

$$\mathcal{O} = \text{Softmax}(\mathcal{A})|_{\text{row}}\mathcal{F}_{audio} = \mathcal{P}\mathcal{F}_{audio} = \mathbf{1}_{N \times 1}\mathcal{F}_{audio} = \begin{bmatrix} 1 \\ 1 \\ \vdots \\ 1 \end{bmatrix} \mathcal{F}_{audio} = \begin{bmatrix} \mathcal{F}_{audio} \\ \mathcal{F}_{audio} \\ \vdots \\ \mathcal{F}_{audio} \end{bmatrix}. \tag{15}$$

The attention dissipation phenomenon shows that cross-attention with visual features such as query and audio as key and value turns out to be a simple replication of audio signals. It goes against our original intent of modality fusion.

### A.1.2 CODE IMPLEMENTATION

To make a fully comprehensive understanding of attention dissipation, we provide a PyTorch-like pseudo-code for easy verification and implementation of cross-attention dissipation. Algorithm 1 provides the pseudo-code of attention dissipation in the AVS task. For the current frame, we calculate the attention matrix with the use of visual features as query and audio as key and value.

---

**Algorithm 1** Pseudo-code of Attention Dissipation in a PyTorch-like style.

```python
# image, audio: visual and audio feature
# attn: attention matrix
# out: output of attention

import torch
import torch.nn as nn
import torch.nn.functional as F

def cross_attention(image:torch.Tensor, audio:torch.Tensor):
    """
    :param image: torch.tensor with shape [B, C, H, W]
    :param audio: torch.tensor with shape [B, C]
    :return: fused feature and attention weight
    """

    image = image.flatten(2).transpose(1, 2)
    audio = audio.unsqueeze(1)

    q = image
    k = audio
    v = audio

    attn = torch.matmul(q, k.transpose(1, 2))
    attn = F.softmax(attn, dim=-1)
    out = torch.matmul(attn, v)

    return out,attn
```

---

## A.2 EXPERIMENTS

### A.2.1 EXPERIMENTAL DETAILS

**Dataset.** We evaluate our method on the AVSBench dataset (Zhou et al., 2022; 2023), which is composed of AVSBench-Object and AVSBench-Semantic. AVSBench-Object is designed for audio-visual segmentation tasks with pixel-level annotations. Videos are sourced from YouTube, cropped into 5 seconds, and sampled at one frame per second to compose the image data. There are two subsets in AVSBench-Object: single sound source segmentation (S4) subset and multiple sound source segmentation (MS3) subset. The S4 subset contains 4,932 videos: 3,452 for training, 740 for validation and 740 for testing. The labels contain 23 categories, including humans, vehicles, animals and kinds of instruments. Note that annotations in S4 training set is only given in the first frame. Meanwhile, MS3 subset is composed of multiple sound sources, including 424 videos, 286 for training, 64 for validation and 64 for testing. MS3 shares the same categories as S4.

**Implementation Details.** During training, we use the original image size as 224×224. We apply horizontal flipping on S4 and MS3 for data augmentation. Since the S4 sub-set only contains annotations on the first frame in the training split, we only use the first frame to provide supervision. We use the AdamW optimizer and a polynomial learning rate decay with power = 0.9. On S4 and MS3, the learning rate is set to 0.0005, and on AVSS, it is set to 0.0001. Following previous practice Gao et al. (2024), we train MS3 for 60 epochs since it is relatively small, while the S4 and AVSS subsets are trained for 30 epochs. Batch size is set to 16 for S4 and MS3 and 8 for AVSS. We adopt two ResNet He et al. (2016) backbones (ResNet-50 and ResNet-18) for the segmentation network. For the audio backbones, we use VGGish Hershey et al. (2017) frozen during the training. The prompt query generator (PQG) receives the feature from the audio backbone as prompt. The number of queries is set to 16, and the number of layers is set to 3. At the output end, the audio feature prompt is discarded. The transformer decoder is adopted from Multi-Scale Deformable (MSDeform) attention Zhu et al. (2020). The first two attention blocks are replaced by convolution to form ELF decoder. Convolution blocks are attached with residual connection and LayerNorm Yu et al. (2024). As for the segmentation loss, on S4 and MS3, we set $\lambda_{\text{IoU}} = 1.8$ and on AVSS $\lambda_{\text{IoU}} = 1.0$ with

$\lambda_{\text{Dice}} = 1.0$ and $\lambda_{\text{aux}} = 0.1$. For inference, since the end-to-end real-time scenario does not support inferring on a bunch of frames (because we want to segment one image at a time on the device), the latency of all models is measured under one single frame, that is, $T = 1$. Nevertheless, some of the methods employ temporal information within multiple frames, which would be lost in a single frame scenario; we still keep their performance the same for comparison.

**Evaluation Metrics.** Following Zhou et al. (2022), we adopt Jaccard index $\mathcal{J}$ and F-score $\mathcal{F}$ to evaluate. $\mathcal{J}$ indicates the mean intersection over union (mIoU) Everingham et al. (2015) between segmentation prediction and ground truth. $\mathcal{F}$ measures the precision and recall by $\mathcal{F} = \frac{(1+\beta^2 \times \text{precision} \times \text{recall})}{\beta^2 \times \text{precision} + \text{recall}}$, where $\beta^2 = 0.3$.

It is important to emphasize that although other methods are evaluated in default AVS settings, that is, with $T$ frames at a time, some of them may show a slight decay because of the absence of temporal information and the appearance of attention dissipation in real-time AVS. But AVESFormer is entirely evaluated under real-time AVS, and hold the same performance in default AVS setting.

### A.2.2 More Results

**Different Backbone.** We provide additional results with another commonly used backbone PVT-v2 (Wang et al., 2022). Results are shown in the following table. With larger scale and more parameters, PVT-v2 gains more performance. However, the inference time of PVT-v2 accounts for a significant proportion up to 86.3% of the whole network. It indicates that the model spends too much time merely on PVT-v2 backbone, while the rest of the network takes 6ms or so. Also, the slight performance improvement of PVT-v2 comes at the cost of nearly 7x inference latency, which is not really efficient. In comparison, ResNet backbones show nice property in the trade-off between performance and inference speed. As a result, we choose ResNet as a more suitable architecture for real-time applications rather than PVT-v2.

Table 9: Performance of different backbones.

| Backbone | S4 | | MS3 | | AVSS | | Latency | Backbone Latency |
| --- | --- | --- | --- | --- | --- | --- | --- | --- |
| | $\mathcal{J}$ | $\mathcal{F}$ | $\mathcal{J}$ | $\mathcal{F}$ | $\mathcal{J}$ | $\mathcal{F}$ | (ms) | (ms) |
| PVT-v2 | 80.5 | 89.2 | 59.5 | 72.3 | 32.9 | 38.5 | 43.8 | 37.8 |
| ResNet50 | 79.9 | 89.1 | 57.9 | 68.7 | 31.2 | 36.8 | 12.0 | 5.5 |
| ResNet18 | 77.3 | 87.5 | 55.0 | 65.1 | 26.3 | 31.8 | 8.8 | 2.4 |

### A.3 Qualitative analysis

### A.3.1 Results Visualization

We present additional visualization results for the paper, alongside AVSBench Zhou et al. (2022), AVSegFormer Gao et al. (2024) and our model on AVSBench-Object Zhou et al. (2022) and AVSBench-Semantic Zhou et al. (2023) with ResNet-50 He et al. (2016) backbone, as depicted in Figure. 11, Figure. 12, and Figure. 13. We demonstrate that AVESFormer efficiently presents a more fine-grained prediction and a more accurate audio-visual corresponding capability to the segmentation of objects in the scene compared to previous methods.

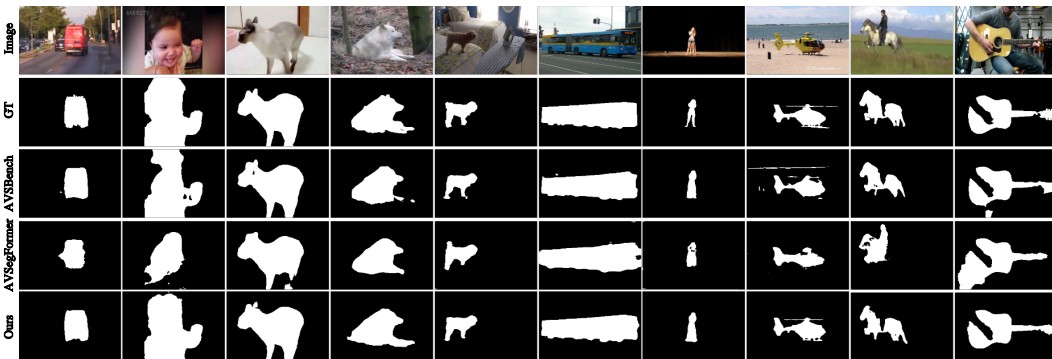

Figure 11: Qualitative audio-visual segmentation results on AVSBench-Object S4 sub-set Zhou et al. (2023) by TPAVI Zhou et al. (2022), AVSegFormer Gao et al. (2024), and AVESFormer. Each row represents the raw image, ground truth or different methods. Each column represents various data samples.

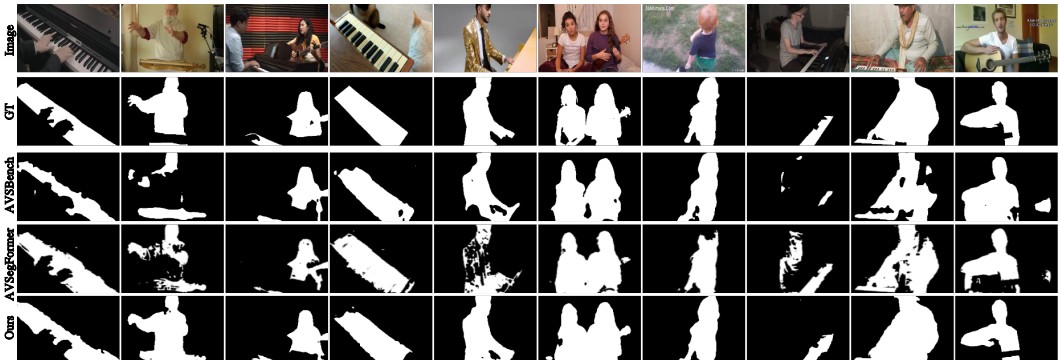

Figure 12: Qualitative audio-visual segmentation results on AVSBench-Object MS3 sub-set Zhou et al. (2023) by TPAVI Zhou et al. (2022), AVSegFormer Gao et al. (2024), and AVESFormer. Each row represents the raw image, ground truth or different methods. Each column represents various data samples.

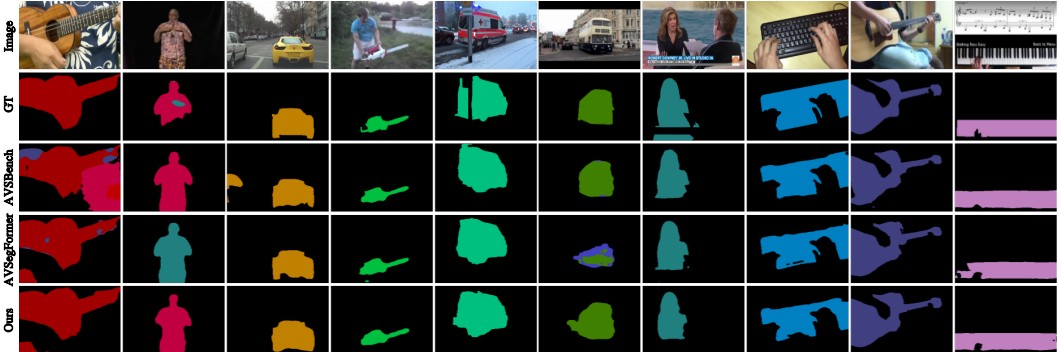

Figure 13: Qualitative audio-visual segmentation results on AVSBench-Semantics Zhou et al. (2023) by TPAVI Zhou et al. (2022), AVSegFormer Gao et al. (2024), and AVESFormer. Each row represents the raw image, ground truth or different methods. Each column represents various data samples.

