# OpenReview forum: "AVESFormer: Efficient Transformer Design for Real-Time Audio-Visual Segmentation"
_ICLR.cc/2025/Conference — Submitted to ICLR 2025_

### Official Review · Reviewer_jhmt · 2024-10-21

**Soundness:** 3
**Presentation:** 3
**Contribution:** 3
**Rating:** 5
**Confidence:** 5

**Summary:**

The authors proposed AVESFormer, an efficient framework for audio-visual segmentation tasks. By solving two issues of using cross-attention as attention dissipation and narrow attention patterns, they can effectively improve the utilization of cross-attention between audio and visual. AVESFormer improves attention utilization, improves the performance as well as reduces the running time.

**Strengths:**

- The paper is well-written and easy to follow.

- The authors offer comprehensive ablation studies, which aid in comprehending the various design decisions involved in solving cross-attention issues.

- The proposed method shows a strong performance improvement as well as a reduction in the inference time.

- Sufficient comparison with SOTA methods

**Weaknesses:**

- Some typo: Table 3: lartency -> latency

- Regarding the PQG, what is the result if we set the number of queries to the number of objects in the category?

- How much of the flops and memory that AVESFormer save from replacing the early attention with convolution?

- Does the number of stages in the decoder affect the performance?

- Missing references [1], [2], and [3] in Table 1

[1] CVPR2024: Audio-Visual Segmentation via Unlabeled Frame Exploitation

[2] Referred by multi-modality: A unified temporal transformer for video object segmentation

[3] Stepping Stones: A Progressive Training Strategy for Audio-Visual Semantic Segmentation

**Questions:**

see Weaknesses

---

> ### Author Response · Authors · 2024-11-25
> **Rebuttal to Reviewer jhmt**
>
> Dear Reviewer jhmt:
>
> We greatly appreciate your insightful and constructive suggestions, which have significantly improved our paper. Let us address your concerns point by point:
>
> 1. For W.1, we apologize for the grammatical errors in our paper and any discomfort they may have caused while reading. We have uploaded a revised version that has been carefully proofread to address these minor issues. We want to assure you that these errors are limited to the writing and do not affect the accuracy of our experiments or conclusions.
>
> 2. For W.2, we have conducted additional ablation experiments on the number of audio queries as per your request. The AVSBench dataset's AVSS sub-task is a multi-class semantic segmentation task. This dataset includes 71 different categories (including the background). We set the number of audio queries to 71 and conducted the experiments as follows:
>
>    | Number of queries | mIoU (S4)   | F score (S4) | mIoU (MS3)  | F score (MS3) |
>    | ----------------- | ----------- | ------------ | ----------- | ------------- |
>    | 8                 | 79.3        | 88.9         | 55.8        | 66.0          |
>    | 16                | 79.9        | 89.1         | 57.9        | 68.1          |
>    | 32                | 79.4        | 88.9         | 56.2        | 66.6          |
>    | 64                | 79.1        | 88.9         | 55.8        | 67.0          |
>    | **71**         | **79.2** | **88.9**  | **56.0** | **67.2**   |
>    | 128               | 79.0        | 88.8         | 56.0        | 67.4          |
>    | 256               | 79.3        | 89.0         | 57.3        | 67.8          |
>
>    We observe that when the number of audio queries is equal to the number of categories, there was no significant improvement in the model's performance. This may be attributed to the similarities and overlaps in audio features across different categories (like laughter, singing & speech), suggesting that a smaller number of queries can effectively represent these audio characteristics.
>
> 3. For W.3, 2e present the corresponding flops and memory usage of AVESFormer with different configurations of the ELF decoder in the table below. Although there is a minor difference in flops and memory between the two, our decoder can significantly improve latency.
>
>    | decoder          | #Params | fps   | latency (ms) | GFLOPs | GPU memory (MB) |
>    | ---------------- | ------- | ----- | ------------ | ------ | --------------- |
>    | ELF              | 127     |  83.5 | 12.0         | 40.3   | 1178            |
>    | pure transformer | 128     | 67.1 | 14.9         | 43.9   | 1210            |
>
> 4. For W.4, we have conducted additional ablation experiments on the number of decoder layers as per your request.  For easier observation, we conducted the following experiments with models having different numbers of decoder layers under the pure Transformer decoder structure as follows：
>
>    | Number of stages | mIoU (S4) | F score (S4) | mIoU (MS3) | F score (MS3) |
>    | ---------------- | --------- | ------------ | ---------- | ------------- |
>    | 6                | 77.3      | 87.6         | 56.2       | 66.6          |
>    | 5                | 75.7      | 87.1         | 52.6       | 65.7          |
>    | 4                | 74.4      | 86.9         | 51.4       | 65.9          |
>    | 3                | 73.0      | 86.7         | 50.5       | 66.2          |
>    | 2                | 71.0      | 86.5         | 47.0       | 65.9          |
>    | 1                | 66.7      | 86.2         | 33.1       | 64.9          |
>
>    The table demonstrates the model's performance at different numbers of decoder layers. As the number of decoder layers decreases, there is a certain degree of performance degradation.
>
> 5. For W5, we realize we have missed these important works that have been really influential in the development of audio-visual segmentation. In our updated version, we've made sure to include their findings.  We notice that reference [3] uses a more advanced Transformer backbone (Swin-B), which might make the comparison of results a bit uneven. To keep things clear and fair, we've shown the relevant performance in the appendix of our AVESFormer with PVT-v2 for further comparison.

---

> > ### Comment · Reviewer_jhmt · 2024-11-26
> >
> > Thank you for answering.
> >
> > 1. Regarding the audio queries, based on the results, it seems like with 256 queries the results become better than 128 and 71 as well, contrasting your assumption on that **"This may be attributed to the similarities and overlaps in audio features across different categories (like laughter, singing & speech), suggesting that a smaller number of queries can effectively represent these audio characteristics."**
> >
> > 2. It's quite weird that ELF has fewer parameters, FLOPs, requires less memory, and has low latency, but also has significantly lower FPS.
> >
> > 3. It is interesting that reducing the stage of the decoder also reduces the performance, but why does that happen?
> >
> > At this state of the paper, I'll keep my initial score.

---

### Official Review · Reviewer_g4aY · 2024-10-31

**Soundness:** 3
**Presentation:** 3
**Contribution:** 3
**Rating:** 6
**Confidence:** 4

**Summary:**

The paper presents AVESFormer, a transformer-based model for real-time audio-visual segmentation AVSAVS. It tackles two main issues in existing AVS models: attention dissipation and limited attention patterns in early decoders. AVESFormer uses a Prompt Query Generator PQGPQG to improve cross-attention and an Early Focus ELFELF decoder that integrates convolutional operations for efficient local feature extraction, reducing computational costs. Experiments show that AVESFormer addresses cross-attention problems, enhances attention utilization, and outperforms previous state-of-the-art models in both performance and speed. Additionally, the paper analyzes attention dissipation and the shortcomings of standard transformer decoders in real-time AVS.

**Strengths:**

A key innovation is the Prompt Query Generator PQGPQG, which corrects cross-attention behavior and mitigates attention dissipation, enhancing segmentation accuracy and efficiency. Additionally, the Early Focus ELFELF decoder incorporates convolutional operations for local feature extraction, reducing computational demands by replacing attention operations in early transformer stages. The model achieves state-of-the-art performance, surpassing previous models on metrics like the Jaccard index and F-score across datasets such as S4, MS3, and AVSS.

**Weaknesses:**

One concern is the significant parameter count of the audio backbone, Vggish, which limits deployment on mobile devices, suggesting future work could optimize this component. Additionally, the model currently ignores temporal information from multiple frames, and incorporating this data could enhance its ability to track moving objects. There is also a need to test the model's generalization to unseen datasets beyond AVSBench to evaluate its robustness in diverse scenarios. While qualitative analyses of attention maps are presented, a quantitative investigation could offer deeper insights into the model's attention mechanisms. Furthermore, comparing AVESFormer with non-transformer models would provide a broader context for its performance. The paper could also benefit from a more detailed discussion on the impacts of model complexity on real-time performance and ethical considerations regarding its application.

**Questions:**

The authors should consider optimizing the audio backbone, Vggish, which currently comprises about 60% of the model parameters, to enable better deployment on mobile devices. Next, the authors are encouraged to explore how incorporating temporal information might influence model performance and inference speed. Additionally, evaluating AVESFormer on a broader range of datasets beyond AVSBench would provide insights into its generalization capabilities. The paper currently lacks a detailed quantitative analysis of attention maps, so including metrics to evaluate the focus and spread of attention could enhance understanding of the model's behavior.

---

> ### Author Response · Authors · 2024-11-25
> **Rebuttal to Reviewer g4aY**
>
> Dear Reviewer g4aY:
>
> We appreciate your effort in reviewing our article and your valuable feedback. We would like to address your concerns as follows.
>
> Your concerns and questions are very reasonable, and they highlight what we also recognize as limitations and pain points.
>
> * For the huge Vggish, given that most existing methods use this backbone, in order to ensure a fair comparison, we have to use the same network as our audio backbone as well. We believe that in future work, exploring how to compress this network to maintain performance while reducing the parameter overhead to meet the demands of mobile devices will be a worthwhile direction to pursue.
>
> * For the temporal information, our network, in its current design, indeed does not utilize temporal information; each frame of data is processed independently without any inter-frame association. We also believe that incorporating temporal information could enhance the network's performance. We are confident that in our future work, we will conduct a more in-depth investigation into this aspect.
>
> * For more datasets, our experiments are also conducted on a new AVS dataset VPO by Ref1. We report the results compared with available methods under ResNet50 backbone:
>
> | Methods           | VPO (SS) mIoU | VPO (SS) F score | VPO (MS) mIoU | VPO (MS) F score |
> | ----------------- | ------------- | ---------------- | ------------- | ---------------- |
> | AVSBench          | 52.8          | 69.5             | 54.3          | 72.0             |
> | AVSegFormer       | 57.6          | 73.0             | 58.3          | 74.3             |
> | AVESFormer (ours) | 59.7          | 76.2             | 59.6          | 75.3             |
>
> * For quantitative analysis of attention maps, we measure the attention entropy of each decoder stage by:
>   $$
>   H(A)=-\sum_i p_i\log(p_i)
>   $$
>   A higher entropy indicates a boarder attention heatmap, while a lower one means a more narrow heatmap focusing on local regions. The results is shown in the table below:
>
>   | Num of decoder stage | attention entropy |
>   | -------------------- | ----------------- |
>   | 1                    | 1.67              |
>   | 2                    | 1.05              |
>   | 3                    | 0.76              |
>   | 4                    | 0.54              |
>   | 5                    | 0.51              |
>   | 6                    | 0.44              |
>
> * For the comparison of non-transformer methods, we have demonstrated the comparison with various popular AVS models from non-transformer and transformer architectures in Table 1, providing a comprehensive analysis on its performance.
>
> * For ethical consideration, it's essential to address a variety of ethical considerations to ensure that the technology is developed and used in ways that are respectful, fair, and accountable. First, for privacy and data consent, Audio-visual data often contain sensitive personal information, such as people’s voices, faces, or behaviors. It is essential to obtain informed consent from individuals whose data is being used for training or testing purposes. Second, for bias and fairness, Audio-visual segmentation systems may exhibit biases depending on the data used for training. Third, for impact on vulnerable populations, Audio-visual segmentation technologies can be misused to track or exploit vulnerable populations (e.g., children, marginalized groups).
>
> Ref1: Unraveling Instance Associations: A Closer Look for Audio-Visual Segmentation. CVPR 2024

---

> > ### Comment · Reviewer_g4aY · 2024-12-03
> >
> > Thanks for the detailed rebuttal comment. I will maintain my current rating.

---

### Official Review · Reviewer_m7g3 · 2024-11-01

**Soundness:** 3
**Presentation:** 3
**Contribution:** 2
**Rating:** 5
**Confidence:** 4

**Summary:**

The paper introduces AVESFormer, a transformer model designed for real-time audio-visual segmentation (AVS). It addresses attention dissipation and narrow attention patterns in AVS by implementing a Prompt Query Generator (PQG) and Early Focus (ELF) Decoder. These components aim to improve cross-attention efficiency and reduce computational costs. AVESFormer shows competitive performance on AVSBench with noted speed improvements.

**Strengths:**

1. The paper identifies critical issues in existing audio-visual segmentation (AVS) models, such as attention dissipation and narrow attention patterns, which are valid challenges in the field.

2. The proposed AVESFormer model aims to address efficiency in real-time AVS tasks, which could be valuable for applications needing low-latency responses.

3. The model demonstrates competitive performance on standard benchmarks with a noted improvement in latency, supporting claims of efficiency gains.

**Weaknesses:**

While the issues identified in attention mechanisms are relevant, the approach—specifically the Prompt Query Generator and Early Focus Decoder—largely leverages incremental modifications rather than substantial innovations. Many elements, such as convolutional layers for early-stage feature extraction, are already established techniques in efficient transformers.

**Questions:**

See weaknesses

---

> ### Author Response · Authors · 2024-11-25
> **Rebuttal to Reviewer m7g3**
>
> Dear Reviewer m7g3:
>
> We appreciate your effort in reviewing our article and your valuable feedback. We would like to address your concerns as follows.
>
> We understand that, to some extent, our work might appear simpler and thus be seen as incremental modifications. However, we must emphasize that compared to designing complex, intricate, and redundant modules and mechanisms, trying to figure out how and why audio-visual segmentation model works or fails holds greater and more valuable significance.
>
> Our work provides a deep analysis of the long-standing failure of cross-attention in the previous AVSegFormer (CRA) and the appearance of plain attention heatmaps in the CAVP paper (Figure 11 (b) in their appendix). We reveal the attention dissipation issue present in the cross-attention fusion stage, which holds significant guiding implications for future work on audio-visual fusion. Similarly, it is precisely because we delve into the essence of the critical fusion problem that our proposed method can achieve sufficient performance improvements in a simple form. Also, it is through our essential analysis that adopting a simple approach can significantly boost the performance of audio-visual segmentation while saving substantial additional computational costs.

---

> > ### Comment · Reviewer_m7g3 · 2024-11-25
> >
> > I still think that the proposed improvements are too loosely related to the audio, so I'll maintain my score.

---

### Official Review · Reviewer_6kJk · 2024-11-01

**Soundness:** 2
**Presentation:** 2
**Contribution:** 2
**Rating:** 5
**Confidence:** 4

**Summary:**

This paper finds two primary challenges in existing audio-visual segmentation models, namely attention dissipation caused by anomalous attention weights after Softmax over limited frames, and narrow attention patterns in shallow decoder stages leading to inefficient utilization of attention resources. An AVESFormer is then presented as the first real-time audio-visual efficient segmentation transformer. Particularly, the proposed method leverages a prompt query generator to rectify cross-attention behavior, and an early focus decoder to enhance efficiency. Extensive experiments demonstrate superiority of AVESFormer in mitigating cross-attention issues and achieving a trade-off between model effectiveness and efficiency.

**Strengths:**

1. Solving AVS via attention analysis is reasonable given the multimodal nature of AVS.
2. The attention dissipation issue seems interesting given the current QKV setting.
3. The experiments are comprehensive.

**Weaknesses:**

1. The two observations which motivate the proposed method are not clear explained (see details in the Questions section).
2. The setting of the paper for real time AVS should also be verified, especially it should be compared to naive settings, e.g. defining the rest of the frames as an empty token. Most importantly, as temporal information is totally discarded, more analysis is needed on how the proposed method achieve smooth real time AVS.
3. Comparison with other attention based techniques is needed (Ref 1 in the Questions section.)

**Questions:**

1. Line 53-76 tries to explain the issues of exiting AVS method. Although it's reasonable that effective multimodal fusion should be critical for AVS, it's not clear why this claim hold: "attention variants generally do not exhibit the same expressive capacity as the default mechanism" (line 75). Further explanation is needed.
2. In Fig. 2, the paper illustrates of attention dissipation. How the audio token is generated? Is this only a one case issue or does it happen all the time in all types of cross attention based AVS? More details are needed to verify the existence of the attention dissipation issue, especially when audio is defined as both key and value in the current cross attention setting. Moreover, where does the cross attention AVS model in equation 1 come from? Citation(s) are needed if it comes from existing literatures, or explanations are needed to explain why QKV is defied in the current way (compared to Ref1, Ref2, Ref3).
3. "Attention maps at early decoder stages tends to capture short-term local correlation features, leading to undesired low utilization of attention" (line 89-91) is used to explain the "narrow attention" issue, which is also not very clear. Why call it "narrow"?
4. In line 123, it's not clear why the existing AVS methods fail to work on the "single frame image + audio" setting.
5. what is the audio backbone in line 184?
6. F_audio=k=v, not clear (line 194). Also, audio feature dimension is D in line 185 and c in line191. It's better to use consistent symbols.
7. The prompt query generator (sec. 4.1) augments the audio feature. How does it bring extra useful information given the augmentation is based on the raw audio feature?
8. The early focus decoder is used to solve the "narrow attention" issue. However, it's not clear how eq. 6 and eq. 7 combined can solve the narrow attention issue.
9. In Table 1, why Ref1 is not compared?
10. In Table 4 and Table 5, it seems the proposed two strategies work better for MS3 dataset, explanations are needed.

Ref1: Unraveling Instance Associations: A Closer Look for Audio-Visual Segmentation. CVPR 2024

Ref2: AVSegFormer: Audio-Visual Segmentation with Transformer. AAAI 2024

Ref3: Audio–visual segmentation. ECCV 2022

---

> ### Author Response · Authors · 2024-11-25
> **Rebuttal to 6kjk**
>
> Dear Reviewer 6kjk:
>
> We appreciate your effort in reviewing our article and your valuable feedback. We would like to address your concerns as follows.
>
> 1. For Q.1, this is the analysis of the research we've cited in that paragraph. Specifically, softmax attention possesses superior properties for measuring the probabilistic relationships between tokens involved in the attention mechanism. Other attention variants, while having lower computational costs, do not enjoy the same probabilistic superiority. For detailed conclusions, please refer to reference 1.
>
> 2. For Q.2, Q.4, Q.6, audio token is provided by audio backbone. It's a case where the k and v are 1-d vectors and can be found in:
>
>    * CRA in AVSegFormer,
>    * Query generator in AVSegFormer,
>    * Figure 11(b) in the Ref1(CAVP),
>    * The Temporal A-to-V Fusion of CATR,
>
>    and so on. As proved by our preliminary, attention map under 1-d k and v leads to an all-1 matrix. Therefore the model could not figure out where to attend.
>
>    For why QKV is defined in current way, to determine which sound-emitting object a visual patch represents, we need to use it as the query, with the audio serving as the key and value. By querying the keys with this query, we can identify the corresponding values, thus determining the sound-emitting object at the current position. Therefore, in the QKV setup, the visual patches should serve as the query (Q), while the audio information should serve as the key (K) and value (V).
>
>    For the misleading alphabet, we apologize for the small errors in our paper and any discomfort they may have caused while reading. We have uploaded a revised version that has been carefully proofread to address these minor issues. We want to assure you that these errors are limited to the writing and do not affect the accuracy of our experiments or conclusions.
>
> 3. For Q.3 and Q.8, "narrow" means for a certain audio token, after undergoing the attention mechanism, it exhibits higher response and attention scores in a very small and localized region, while the rest of the area shows much lower scores. And ELF decoder is not used to solve it. This is a strategy proposed to save computational resources based on observations of current cross-attention behavior, where the original attention map has not been specifically altered.
>
> 4. For Q.5, the audio backbone is used to extract audio features from mel-spectrum of a piece of audio. Vggish is commonly used by almost all AVS methods.
>
> 5. For Q.7, as shown in Figure 9,  for a given frame, certain audio queries attend to the corresponding sounding object, while others may focus on the background. Each audio query captures distinct semantic features: some attend to specific parts of the sounding object, while others capture the entire object. Across different frames, queries adapt by attending to different objects. For instance, a query might focus on the sounding object in one frame but shift attention to the background in a different context.
>
> 6. For Q.9, as described by the appendix of Ref1 and its open-sourced code, Ref1 is trained on 512x512 resolution, while most of the AVS models are trained and evaluated on 224x224 resolution. We did not include it, considering the need for a consistent training resolution. However, this work's main contribution lies in its novel training strategy, which uses a relatively common network architecture. This makes our innovation points orthogonal. We believe that better results can be achieved by combining the methods from both of our works.
>
> 7. For Q.10, from the perspective of tasks, S4 shows quite a different property from MS3. In the single source subset S4, we find that sounding objects are commonly at the saliency position. AVS models could still segment correct objects as a kind of saliency object segmentation, even without the help of audio (See Table 4. w/o QG). That is, audio-visual correspondence plays a minor role in the S4 task. While in the MS3 subset, multiple objects appear in a single image, some may be sounding while some may not, which requires better audio-visual distinguishing ability than S4. With attention dissipation, it is easier to distinguish multiple-sounding objects, which further harms the performance. As we tackle attention dissipation with PQG, the model gains the distinguishing ability and shows satisfied performance.

---

### Meta-Review · Area_Chair_U83J · 2024-12-19

**Metareview:**

This work received three negative ratings (borderline reject) and one positive rating (borderline accept). The reviewers rate negative scores due to the limited novelty, lack of high-level explanation of the motivation and contradictory experimental results raised by reviewer jhmt. The authors also did not further address the comments of the reviewer jhmt. AC checked the comments of the reviewers and thus made the final decision.

**Additional Comments On Reviewer Discussion:**

The novelty and the high-level motivations are the most concerned parts of this work. Reviewers have pointed out the experiments contradict the assumption of the paper and some phenomenon cannot be explained. Also reviewers raised that the innovations are not substantial. Many elements are already established in efficient transformers.

---

### Decision · Program_Chairs · 2025-01-22

Reject